# Mitotic fidelity requires transgenerational action of a testis-restricted HP1

Mia T Levine[1†], Helen M Vander Wende[1], Harmit S Malik[1,2]*

[1]Division of Basic Sciences, Fred Hutchinson Cancer Research Center, Seattle, United States; [2]Howard Hughes Medical Institute, Fred Hutchinson Cancer Research Center, Seattle, United States

**Abstract** Sperm-packaged DNA must undergo extensive reorganization to ensure its timely participation in embryonic mitosis. Whereas maternal control over this remodeling is well described, paternal contributions are virtually unknown. In this study, we show that *Drosophila melanogaster* males lacking *Heterochromatin Protein 1E* (*HP1E*) sire inviable embryos that undergo catastrophic mitosis. In these embryos, the paternal genome fails to condense and resolve into sister chromatids in synchrony with the maternal genome. This delay leads to a failure of paternal chromosomes, particularly the heterochromatin-rich sex chromosomes, to separate on the first mitotic spindle. Remarkably, HP1E is not inherited on mature sperm chromatin. Instead, HP1E primes paternal chromosomes during spermatogenesis to ensure faithful segregation post-fertilization. This transgenerational effect suggests that maternal control is necessary but not sufficient for transforming sperm DNA into a mitotically competent pronucleus. Instead, paternal action during spermiogenesis exerts post-fertilization control to ensure faithful chromosome segregation in the embryo.

*For correspondence: hsmalik@fhcrc.org

Present address: †Department of Biology, University of Pennsylvania, Philadelphia, United States

Competing interests: The authors declare that no competing interests exist.

## Introduction

Faithful chromosome segregation requires careful orchestration of chromosomal condensation, alignment, and movement of mitotic chromosomes during every eukaryotic cell division (*Rhind and Russell, 2012*). The very first embryonic mitosis in animals requires additional synchronization. Paternally and maternally inherited genomes undergo independent chromatin reorganization and replication prior to mitotic entry. For instance, maternal chromosomes must complete meiosis (*Sen et al., 2013*) and then transition from a meiotic conformation to an interphase-like state in preparation for replication. The sperm-deposited, paternal chromosomes must undergo an even more radical transition from a highly compact, protamine-rich state to a decondensed, histone-rich state before DNA replication (*Braun, 2001*; *Miller et al., 2010*). Despite these divergent requirements to achieve replication- and mitotic-competency, maternal and paternal genomes synchronously enter the first mitosis. Failure to carry out paternal chromosome remodeling in a timely fashion results in paternal genome loss and embryonic inviability (*Loppin et al., 2001*; *McLay and Clarke, 2003*; *Landmann et al., 2009*).

The transition from a protamine-rich sperm nucleus to a competent paternal pronucleus requires the action of numerous maternally deposited proteins in the egg (*McLay and Clarke, 2003*). For instance, paternal genome decondensation post-fertilization requires the integration of histone H3.3, a histone variant deposited by the maternal proteins HIRA (*Loppin et al., 2005a*), CHD1 (*Konev et al., 2007*), and Yemanuclein (*Orsi et al., 2013*). Similarly, maternally-deposited MH/Spartan protein localizes exclusively to the replicating paternal genome and is required for faithful paternal chromosome segregation during the first embryonic division (*Delabaere et al., 2014*). These and other studies demonstrate the essential role of maternally-deposited machinery in

**eLife digest** The genetic information of cells is packaged into structures called chromosomes, which are made up of long strands of DNA that are wrapped around proteins to form a structure called chromatin. The cells of most animals contain two copies of every chromosome, but the egg and sperm cells contain only one copy. This means that when an egg fuses with a sperm cell during fertilization, the resulting 'zygote' will contain two copies of each chromosome—one inherited from the mother, and one from the father. These chromosomes duplicate and divide many times within the developing embryo in a process known as mitosis.

The first division of the zygote is particularly complicated, as the egg and sperm chromosomes must go through extensive—and yet different—chromatin reorganization processes. For instance, paternal DNA is inherited via sperm, where specialized sperm proteins package the DNA more tightly than in the maternal DNA, which is packaged by histone proteins used throughout development. For paternal DNA to participate in mitosis in the embryo, it must first undergo a transition to a histone-packaged state. Despite these differences, both maternal and paternal chromosomes must undergo mitosis at the same time if the zygote is to successfully divide. Although it is known that the egg cell contributes essential proteins that are incorporated into the sperm chromatin to help it reorganize, the importance of paternal proteins in coordinating this process remains poorly understood.

Many members of a family of proteins called Heterochromatin Protein 1 (or HP1 for short) have previously been shown to control chromatin organization in plants and animals. In 2012, researchers found that several HP1 proteins are found only in the testes of the fruit fly species *Drosophila melanogaster*. They predicted that these proteins might help control the reorganization of the paternal chromosomes following fertilization.

Levine et al.—including researchers involved in the 2012 study—have now used genetic and cell-based techniques to show that one member of the HP1 family (called HP1E) ensures that the paternal chromosomes are ready for cell division at the same time as the maternal chromosomes. Male flies that are unable to produce this protein do not have any offspring because, while these flies' sperm can fertilize eggs, the resulting zygotes cannot divide as normal.

Further experiments revealed that HP1E is not inherited through the chromatin of mature sperm, but instead influences the structure of the chromosomes during the final stages of the development of the sperm cells in the fly testes.

This study shows that both maternal and paternal proteins are needed to control how the paternal chromosomes reorganize in fruit fly embryos. Although difficult to discover and decipher, this work re-emphasizes the importance of paternal epigenetic contributions—changes that alter how DNA is read, without changing the DNA sequence itself—for ensuring the viability of resulting offspring. Future work will reveal both the molecular mechanism of this epigenetic transfer of information, as well as why certain *Drosophila* species are able to naturally overcome the loss of the essential HP1E protein.

rendering competent sperm-deposited DNA and ultimately, ensuring faithful paternal genome inheritance.

Is paternal control also necessary for the extensive decondensation and re-condensation of the post-fertilization paternal genome? If so, disruption of such control would manifest as paternal effect lethality (PEL). Unlike male sterility mutants that lack motile sperm, PEL mutants make abundant motile sperm that fertilize eggs efficiently. However, embryos 'fathered' by PEL mutants are inviable. Only a handful of PEL genes have been characterized in animals (*Browning and Strome, 1996*; *Fitch and Wakimoto, 1998*; *Fitch et al., 1998*; *Loppin et al., 2005b*; *Smith and Wakimoto, 2007*; *Gao et al., 2011*; *Seidel et al., 2011*). These encode proteins that mediate sperm release of paternal DNA, sperm centriole inheritance, and paternal chromosome segregation. Only one of these PEL proteins directly localizes to paternal chromosomes; the sperm-inherited K81 protein localizes exclusively to paternal chromosome termini and ensures telomere integrity (*Dubruille et al., 2010*; *Gao et al., 2011*). The maintenance of telomeric epigenetic identity joins a growing list of examples of sperm-to-embryo information transmission via protein or RNA inheritance (e.g., diet: [*Ost et al., 2014*],

stress: [*Rodgers et al., 2013*], embryonic patterning: [*Bayer et al., 2009*], transcriptional competency: [*Hammoud et al., 2010*; *Rando, 2012*; *Ihara et al., 2014*]). Despite our new appreciation of paternal control over epigenetic information transfer, there are no reports of paternal control over the global chromatin reorganization required for synchronous mitosis across paternally and maternally inherited genomes. Indeed, in the absence of any known paternal protein-directed genome remodeling, a model has emerged that maternal proteins might be sufficient for transforming tightly packaged sperm DNA into a fully competent paternal pronucleus.

The notion that maternal control is sufficient to accomplish paternal genome remodeling is challenged by recent findings from the intracellular *Wolbachia* bacterium that infects more than 50% of insect species (*Hilgenboecker et al., 2008*). *Wolbachia*-infected *Drosophila* males mated to uninfected females father embryos that arrest soon after the first zygotic mitosis (*Lassy and Karr, 1996*). Embryonic arrest occurs because paternal genomes enter the first mitosis with unresolved sister chromatids that fail to separate on the mitotic spindle (*Callaini et al., 1997*; *Landmann et al., 2009*). Although the identity of the host factor(s) manipulated by *Wolbachia* to mediate this transgenerational effect is still unknown, what is clear is that pre-fertilization, *Wolbachia* subverts the paternal germline machinery that helps direct global genome remodeling of paternal chromosomes in the embryo. *Wolbachia* action during spermiogenesis leads to paternal-maternal genome asynchrony and ultimately, failure of paternal chromosomes to separate on the first mitotic spindle (*Callaini et al., 1997*; *Landmann et al., 2009*). Despite decades of interest, the molecular basis of paternal control has remained elusive.

To investigate the potential for paternal control over sperm genome remodeling post-fertilization, we took a candidate gene approach, focusing on the Heterochromatin Protein 1 (HP1) proteins that orchestrate genome-wide chromosomal organization in plants, animals, fungi, and some protists (*Lomberk et al., 2006*). HP1 proteins are defined as such by a combination of two domains—a chromodomain that mediates protein-histone interactions and a chromoshadow domain that mediates protein–protein interactions (*Aasland and Stewart, 1995*; *Eissenberg and Elgin, 2000*). The biochemical properties of HP1 members (*Canzio et al., 2014*) support a diversity of chromatin-dependent processes in the soma, including DNA replication (*Pak et al., 1997*; *Schwaiger et al., 2010*), telomere integrity (*Fanti et al., 1998*), and chromosome condensation (*Kellum et al., 1995*).

Recently, we carried out a detailed phylogenomic analysis of the HP1 gene family in *Drosophila* that revealed numerous testis-restricted HP1 proteins (*Levine et al., 2012*). Given the established roles of HP1 proteins (*Eissenberg and Elgin, 2000*; *Lomberk et al., 2006*; *Vermaak and Malik, 2009*; *Canzio et al., 2014*), we posited that these newly discovered male-specific *HP1* genes might represent excellent candidates for encoding chromatin functions specialized for paternal genome organization and remodeling in the early embryo. Using detailed genetic and cytological analyses, here we show that one of these testis-specific HP1 proteins, *Heterochromatin Protein 1E* (HP1E), is essential for priming the paternal genome to enter embryonic mitosis in synchrony with the maternal genome in *D. melanogaster*. Intriguingly, HP1E is able to mediate this priming function transgenerationally i.e., the HP1E protein itself is not epigenetically inherited. We further show that absence of HP1E especially imperils mitotic fidelity of the heterochromatin-rich, paternal sex chromosomes. Thus, our study firmly establishes that both maternal and paternal control are necessary for paternal genome remodeling in the early *Drosophila* embryo.

## Results

### HP1E encodes a spermiogenesis-restricted chromatin protein

The *HP1E* gene is a testis-restricted *Drosophila HP1* paralog born more than 60 million years ago (*Levine et al., 2012*). To investigate the possibility that HP1E acts during chromatin reorganization prior to sperm maturation, we generated transgenic flies that encoded a Flag- or YFP-tagged HP1E fusion protein, driven by the native *HP1E* promoter. In addition, we raised a highly specific polyclonal antibody against HP1E (*Figure 1—figure supplement 1*). All three reagents revealed that HP1E localizes to developing spermatids subsequent to the completion of meiosis II (*Figure 1A–C*, *Figure 1—figure supplements 2, 3, 4*), ruling out a role for HP1E during the pre-meiotic or meiotic phases of spermatogenesis. HP1E signal was generally diffuse across a subset of the chromatin throughout spermiogenesis (*Figure 1A–C*) but disappeared completely at sperm maturation (*Figure 1D*). Native expression of the *HP1E-YFP* fusion transgene confirmed our immunofluorescence

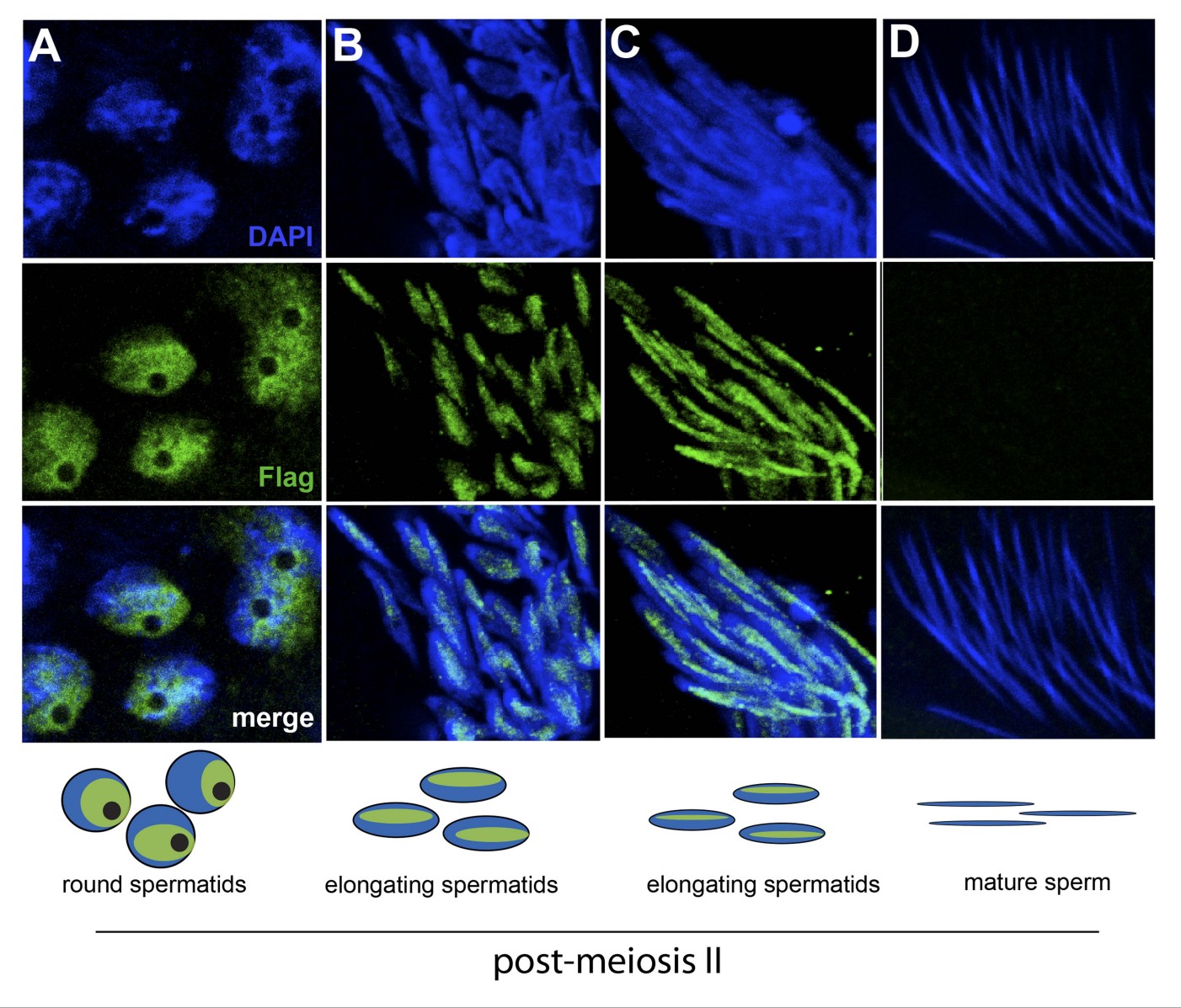

**Figure 1**. HP1E localization in *D. melanogaster* spermatogenesis. We highlight four stages of spermatogenesis in *D. melanogaster* testes: (**A**) round spermatids, (**B**, **C**) elongated spermatids, and (**D**) mature sperm. HP1E localization was visualized with a Flag epitope-tagged HP1E transgene driven by a native promoter. We find that Flag-HP1E protein (green) localizes to the DNA (blue) of post-meiotic spermatids and persists until sperm maturation but is not present on mature sperm.

The following figure supplements are available for figure 1:

**Figure supplement 1**. HP1E antibody is specific.

**Figure supplement 2**. HP1E protein localizes to post-meiotic spermatids.

**Figure supplement 3**. Anti-3xFlag (M2) exhibits no localization to spermatids in a non-Flag tagged HP1E genetic background.

**Figure supplement 4**. Fixed testis expressing *HP1E-YFP* driven by a native promoter recapitulates immunofluorescence results.

results (*Figure 1—figure supplement 4*), implying that HP1E disappearance in late spermiogenesis is not due to antibody inaccessibility in the highly condensed sperm head. These data demonstrate that HP1E localizes to paternal chromosomes only during the radical reorganization of histone-rich chromatin into protamine-rich sperm DNA and disappears once sperm mature.

Many characterized HP1 proteins serve critical heterochromatin organization roles and are often used as markers of canonical heterochromatin in somatic cells (*Eissenberg and Elgin, 2000*). However, heterochromatin organization is poorly defined in post-meiotic developing spermatids (*Dubruille et al., 2010*; *Hennig and Weyrich, 2013*), precluding our ability to ask if HP1E co-localizes with classic bulk heterochromatin markers. We were also unable to directly ascertain HP1E localization to specific heterochromatin loci using chromatin immunoprecipitation based methods (e.g., ChIP-seq). Instead, we adopted an orthogonal approach. We conducted RNA-seq on control and HP1E-depleted testes, reasoning that loss of heterochromatin organizing protein would uniquely perturb the global transcriptional readout from this specialized genome compartment. We found that *HP1E* knockdown during this narrow developmental stage affects the expression of over 700 genes (fdr < 0.05, *Figure 2—source data 1*). Of those 700 genes significantly misregulated upon knockdown, there were very few genes that encode known chromatin-modifying or chromosome-bound proteins; of these, most represent uncharacterized genes. However, one intriguing pattern that does emerge from this dataset is that 100% of significantly misregulated heterochromatin-embedded genes are upregulated when *HP1E* is depleted (*Figure 2*). In comparison, less than 60% of significantly misregulated euchromatin-embedded genes are upregulated. This dichotomy between the effects of *HP1E* depletion on the euchromatic and heterochromatic compartment is highly significant (p < 0.00001) and suggests a direct action of HP1E in the heterochromatic compartment, akin to its closest HP1 relative, HP1A (*Kellum et al., 1995*; *Levine et al., 2012*). Alternatively, one or more of the 700 mis-regulated genes could be responsible for modifying paternal chromatin. However, no obvious candidate genes involved in chromatin modification or binding emerged from the list of mis-regulated genes (*Figure 2—source data 1*). In the absence of direct evidence of heterochromatin localization via cytology or ChIP-seq, we can only tentatively conclude that HP1E acts directly on this genome compartment. In contrast, HP1E unambiguously localizes to chromatin during sperm development.

## *HP1E* is a paternal effect lethal gene in *D. melanogaster*

To investigate *HP1E*'s role in male fertility, we generated *HP1E*-depleted fathers by driving a UAS promoter-hairpin homologous to the *D. melanogaster HP1E* transcript with either an *actin5C*-Gal4 driver (ubiquitous expression) or *vasa*-Gal4 driver (male germline expression). Both drivers efficiently knocked down *HP1E* expression (*Figure 3—figure supplement 1*) and both resulted in highly penetrant male sterility (*Figure 3A*). To rule out off-target effects of RNAi, we engineered a recoded version of *HP1E* in which all synonymous sites were changed but the amino acid sequence was preserved. This recoded, RNAi-resistant *HP1E* transgene (*Figure 3A*, *Figure 3—figure supplement 2*) fully rescued fertility. In parallel, we also generated an *HP1E*-null allele using a TAL-effector nuclease (*Figure 3—figure supplement 3*). We found that *HP1E* knockout males are also completely sterile, and this sterility is also fully reversed by the recoded *HP1E* transgene (*Figure 3—figure supplement 4*). Thus, *HP1E* is required for male fertility in *D. melanogaster*.

Unlike the vast majority of male sterility mutants, HP1E-depleted fathers produce abundant motile sperm that transfer to females, fertilize eggs, and initiate embryogenesis (*Figure 3B*). However, embryos sired by *HP1E*-depleted fathers failed to hatch (0.5% hatch rate). These data demonstrate that *HP1E*-depletion results in PEL i.e., zygotic viability is dependent on father's genotype. Embryos fathered by HP1E-depleted males ('PEL embryos' hereafter) arrested after only a few rounds of zygotic mitosis (*Figure 3C*) and exhibited a chromatin bridge in the very first mitotic division (*Figure 4A*). Aged embryos exhibit increasingly asynchronous nuclear cycling and acute mitotic catastrophe (*Figure 4B*). This gross chromosome segregation defect ultimately results in highly penetrant embryonic lethality.

## HP1E is required for fidelity of paternal chromosome segregation

To gain insight into the mechanism of paternal effect lethality in the PEL embryos, we tracked maternal and paternal DNA dynamics prior to the first telophase. We stained fixed, 0–20 min-old embryos with DAPI and an acetylated histone 4 (AcH4) antibody (*Figure 5A*). AcH4 accumulates

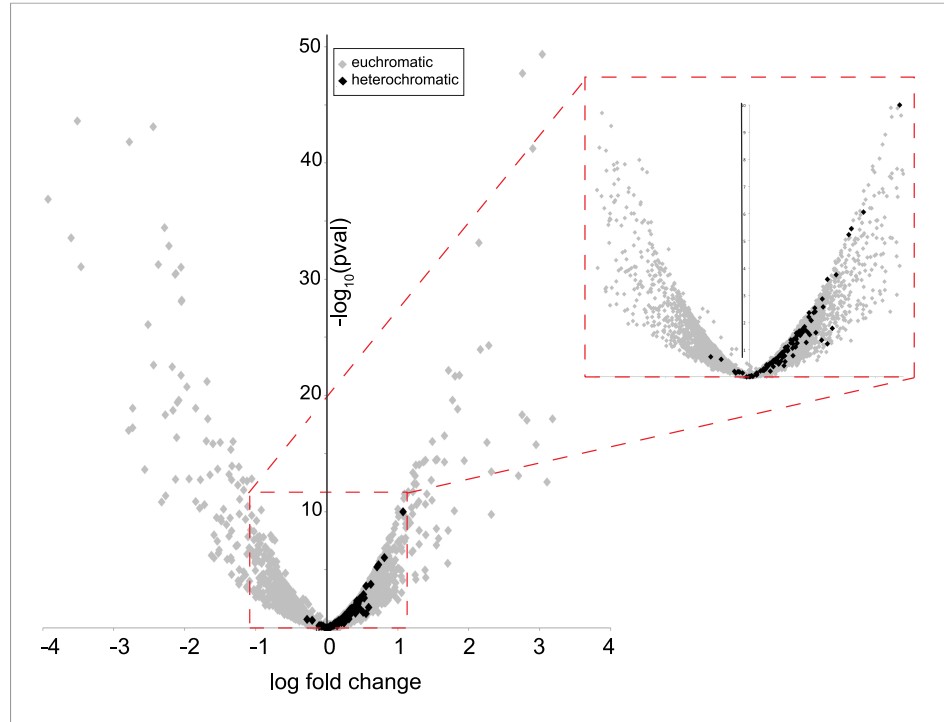

**Figure 2**. Heterochromatin-embedded genes are globally perturbed upon HP1E-depletion. HP1E depletion in testis results directly or indirectly in mis-regulation of hundreds of genes. Volcano plot illustrates the fold up- and down-regulation of euchromatin-embedded genes (gray points) and heterochromatin-embedded genes (black points).

The following source data is available for figure 2:

**Source data 1**. Results of RNA-seq comparisons between testes of wild-type vs HP1E-depleted males, rank-ordered by the false discovery rate.

preferentially on paternal chromatin prior to and during embryonic mitotic cycle 1, allowing us to distinguish paternal from maternal DNA (*Adenot et al., 1997*). PEL embryos revealed no gross defects in maternal chromatin dynamics (*Figure 5A*) and exhibited stereotypical centrosome and spindle morphology (*Figure 5—figure supplement 1*). Furthermore, sperm DNA in PEL embryos underwent the protamine-to-histone transition (*Figure 5—figure supplement 2*), decondensed, migrated toward the maternal pronucleus, and entered into the first mitosis just like in wild-type embryos (*Figure 5A*). Using antibodies against a replication protein (PCNA) and a kinetochore protein (Cenp-C), we also found that both pronuclei recruit replication machinery and initiate kinetochore assembly in PEL embryos (*Figure 5—figure supplements 3, 4*).

The first visible sign of defects in PEL embryos was observed at metaphase. We found that paternal DNA failed to condense synchronously with the maternal chromosomes in PEL embryos (*Figure 5A,B*). To quantify this asymmetry, we adopted a measure of 'circularity' (see 'Materials and methods') of maternal and paternal components across wild-type and PEL embryos (*Figure 5B*). Condensed chromosomes appear as finger-like projections and therefore exhibit circularity close to 0. In contrast, interphase chromosomes appear close to a perfect circle and exhibit circularity close to 1. In wild-type embryos, we found that the ratio of paternal to maternal circularity was equal to one, suggesting that both pronuclei undergo synchronous condensation. In contrast, paternal DNA had twice the circularity of maternal DNA in PEL embryos, suggesting asynchronous condensation due to failure of the paternal genome to compact into resolved chromatids in a timely fashion.

Immediately following this asynchronous metaphase in PEL embryos, we found that the paternal DNA failed to separate on the mitotic spindle. Specifically, a prominent chromatin bridge enriched in paternal chromatin-specific AcH4 appeared in PEL embryos at the first anaphase (*Figure 5A*, arrowhead, n = 20/20). We also observed AcH4 at the poles, suggesting that only a fraction of the

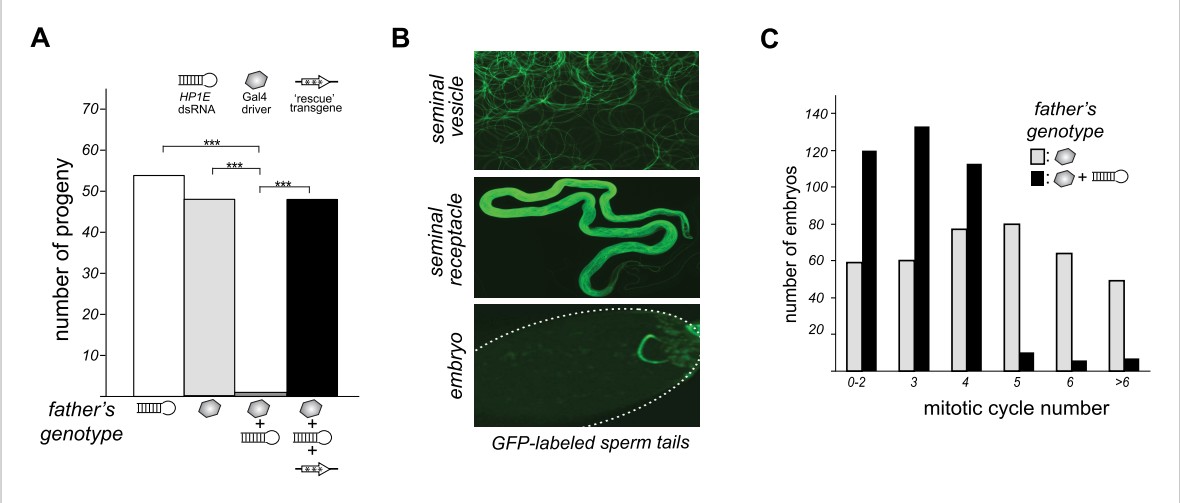

**Figure 3**. *HP1E* is a paternal effect lethal in *D. melanogaster*. (**A**) *HP1E* knockdown via simultaneous presence of both a UAS-driven *HP1E* dsRNA gene and a Gal4 driver results in highly penetrant male sterility. Fertility can be fully restored by an *HP1E* transgene recoded at all synonymous sites, driven by a native *HP1E* promoter ('rescue transgene'). Please refer to *Figure 3—source data 1*. (**B**) HP1E-depleted males produce abundant motile sperm (seminal vesicle), which are efficiently transferred to females (seminal receptacle), and fertilize the egg (embryo). We visualized sperm tails using the *'don juan-GFP'* transgene (29). (**C**) Unlike wild-type embryos (gray), embryos fathered by HP1E-depleted males (black) arrest after 3–4 rounds of nuclear divisions (Mann–Whitney U: p < 0.0001). Embryos were collected in the 5–70 min window post-fertilization. Please refer to *Figure 3—source data 2*.

The following source data and figure supplements are available for figure 3:

**Source data 1**. Number of progeny fathered by males encoding both the UAS-*HP1E* hairpin and the Gal4 driver (24196/A5C) compared to fathers encoding the Gal4 transgene alone (w1118/A5C), hairpin alone (24196/CyO), or both plus the native promoter-driven, *HP1E* recoded transgene (24196/A5C + transgene).

**Source data 2**. Mitotic cycle number (0, 1, 2 etc) of embryos fathered by wild-type males (24196/TM6) or PEL embryos fathered by HP1E-depleted males (24196/A5C) collected during the 75-min window post-oviposition.

**Figure supplement 1**. HP1E knockdown using multiple drivers is efficient.

**Figure supplement 2**. DNA sequence of recoded HP1E transgene.

**Figure supplement 3**. Nucleotide and amino acid sequence of the HP1E mutant.

**Figure supplement 4**. *HP1E* mutant recapitulates male fertility defect.

paternal genome mis-segregates in PEL embryos. Based on these findings, we conclude that defects in paternal chromatin organization are the primary source of mitotic arrest in embryos fathered by HP1E-depleted males. However, our analyses could not formally rule out the possibility that mitotic defects additionally result from maternal chromatin defects, the loss of essential mitotic machinery normally contributed by wild-type fathers (e.g., the centriole), or the deposition of a mitotic 'poison' by the PEL fathers.

To test whether paternal chromatin alone was sufficient to trigger failed mitosis in PEL embryos, we adopted a genetic approach that took advantage of the *D. melanogaster* maternal effect lethal *sesame*[185b] (*Loppin et al., 2000*). In eggs laid by homozygous mutant *sesame* females, the paternal DNA completely fails to de-condense and so does not participate in zygotic mitosis. Instead, the haploid maternal chromosomes undergo mitotic cycling like wild-type diploid embryos until late embryogenesis—long after the PEL mitotic arrest observed in embryos sired by HP1E-depleted males (*Figure 3C*). We crossed the *HP1E*-depleted males to *sesame* females to ask whether bypassing paternal chromatin was sufficient to rescue mitotic cycling. We observed full rescue of nuclear divisions in these crosses (*Figure 6A*); the resulting embryos cycled maternal haploid nuclei identically to those

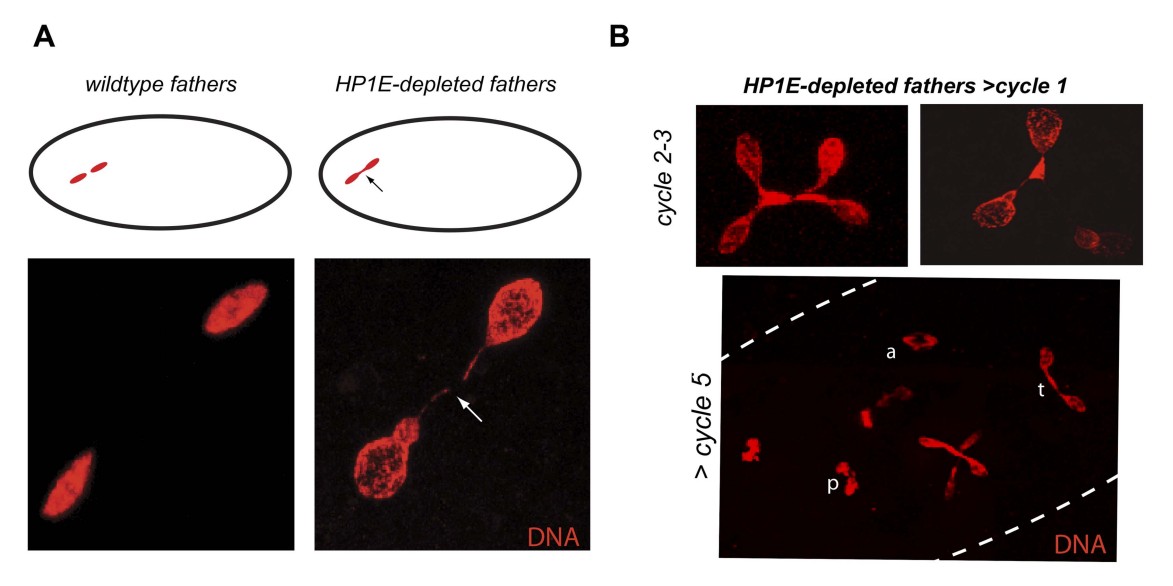

**Figure 4**. HP1E depletion in testis results in failed first embryonic mitosis and later mitotic catastrophe. (**A**) We observed a chromatin bridge (arrow) in the first zygotic telophase in PEL embryos fathered by HP1E-depleted, but not wildtype, males. (**B**) Embryos aged beyond first mitosis exhibit increasingly aberrant nuclear morphology and asynchrony across nuclei (p = prophase, a = anaphase, t = telophase).

'fathered' by a wild-type male beyond embryonic cycle 12 (85% and 82%, respectively, p > 0.2, *Figure 6A*). We therefore conclude that defects in paternal chromatin dynamics are both necessary and sufficient to explain the mitotic arrest in PEL embryos sired by HP1E-depleted males.

When does this paternal chromatin defect arise? We failed to detect HP1E protein cytologically in both wild-type mature sperm head (*Figure 1D*, *Figure 1—figure supplement 3*) and wild-type early embryos (data not shown). Nevertheless, we wanted to formally consider the possibility that low levels of sperm-inherited HP1E might act during early embryogenesis to ensure proper mitosis. If this was the case (as with *spe-11* [*Browning and Strome, 1996*]), the defects we observed arising in PEL embryos could be due to missing HP1E protein in the embryo itself. To test this possibility, we used ectopic over-expression of *HP1E* during oogenesis to maternally deposit HP1E protein into the embryo. This over-expression strategy resulted in robustly detectable HP1E levels in the embryo (*Figure 6B*), whereas embryos laid by control females harbor no detectable HP1E. However, maternally deposited HP1E was unable to rescue the PEL defect associated with sperm from HP1E-depleted males (*Figure 6B*). Consistent with the testis cytology presented in *Figure 1*, these data suggest that it is HP1E action during spermiogenesis, rather than in early embryogenesis, which is responsible for ensuring proper embryonic mitosis. Combined with the successful rescue of mitotic cycling by *sesame-* mothers, we conclude that HP1E primes the paternal genome during spermiogenesis i.e., pre-fertilization, to ensure proper remodeling of the paternal genome in embryos post-fertilization.

## Heterochromatin-rich paternal sex chromosomes are especially vulnerable to HP1E depletion

Our analysis of the anaphase bridges in PEL embryos revealed that only a fraction of the paternal genome appears to be affected by the HP1E depletion (*Figure 5A*). This unusual observation suggested the possibility that all five *D. melanogaster* chromosomes might not be equally dependent on HP1E function. Based on our previous findings that HP1E-depletion led to a global overexpression of heterochromatin-embedded genes (*Figure 2*), we speculated that the paternal chromosomes that encode the longest tracts of heterochromatin might be especially sensitive to HP1E depletion. In the *D. melanogaster* genome, heterochromatic DNA is most abundant on the sex chromosomes (*Celniker and Rubin, 2003*).

To test the possibility that specific chromosomes are enriched in the chromatin bridge, we performed fluorescent in situ hybridization (FISH) analysis on wild-type and PEL embryos using chromosome-specific

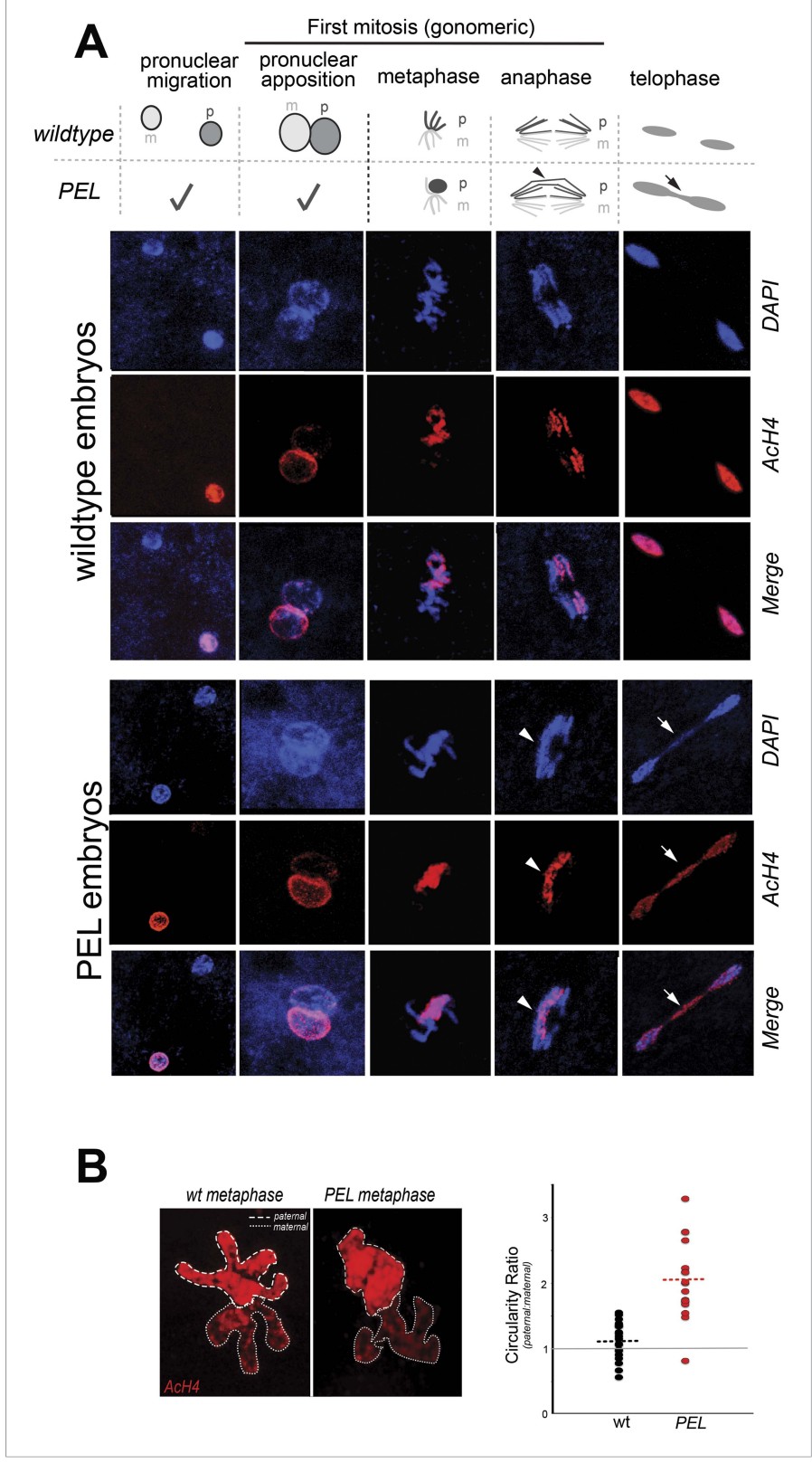

**Figure 5**. HP1E depletion in fathers results in mitotic arrest due to a paternal chromatin defect. (**A**) Paternal chromatin (marked by anti-AcH4 (red)) morphology mirrors maternal chromatin in wild-type embryos but differs in PEL embryos. In both wild-type and PEL embryos, the female and male pronuclei 'migrate' toward each other, 'appose', and then enter mitosis. However, in PEL embryos, metaphase appears asynchronous between maternal

*Figure 5. continued on next page*

*Figure 5. Continued*

and paternal chromatin, an AcH4-enriched chromatin bridge appears in anaphase (arrowhead) and persists at telophase (arrow). (**B**) We calculated a 'circularity ratio' (1 = perfect circle, 0 = starfish) for the first metaphase in wild-type and PEL embryos. We found that the paternal chromatin was significantly more circular i.e., less condensed than maternal chromatin in PEL (red dots) but not wild-type embryos (black dots), (Mann Whitney-U test, $p < 0.0001$). Dotted lines refer to sample means. A circularity ratio of 1 (gray solid line) refers to paternal and maternal chromatin with equivalent circularity.

The following source data and figure supplements are available for figure 5:

**Source data 1**. Independent measurements of 'circularity' of maternal to paternal nuclei at first metaphase in embryos fathered by either wild-type ('wt') or HP1E mutant males ('HP1E').

**Figure supplement 1**. In embryos fathered by both wild-type and HP1E knockdown males, metaphase centrosomes (red, left panel) and spindle (red, right panel) are indistinguishable.

**Figure supplement 2**. Embryos fathered by HP1E knockout males eject protamines.

**Figure supplement 3**. PCNA (replication factor) is recruited to both maternal and paternal pronuclei at apposition in wild-type and PEL embryos.

**Figure supplement 4**. HP1E PEL embryos initiate kinetochore assembly.

satellite probes to all five *D. melanogaster* chromosomes (*Dernburg, 2011*) (*Figure 7A,C*). We found that the paternal Y was trapped as a bridge between nuclei in 94% of the male PEL embryos (*Figure 7B*), whereas the maternal X was never found in the bridge (*Figure 7B*). In female embryos, we found that the (inferred) maternal X-chromosome segregated faithfully while the (inferred) paternal X-chromosome was trapped as a bridge in 60% of embryos. We found that the large autosomes—chromosomes II and III—mis-segregated at only 4% and 15% frequency in male and female embryos, respectively. The dichotomy between sex chromosomes and the large autosomes is highly significant ($p < 0.0001$, *Figure 7D*). Homology between the small autosomal fourth ('dot') and Y-chromosomes precluded us from inferring fourth chromosome mis-segregation frequencies in male embryos. However, in female embryos, we found that the paternal fourth chromosome mis-segregated at 25% frequency, intermediate between the autosomes and sex chromosomes.

Our discovery of sex chromosome enrichment in the telophase bridge suggests that a heterochromatic locus common to the X and Y chromosomes may underlie PEL. The only known repetitive locus exclusive to the X and Y in *D. melanogaster* is the multigene cluster of rDNA genes, which encode the ribosomal RNAs. We hybridized labeled rDNA probes (targeting the IGS sequence) in combination with the Y-satellite probe to wild-type (*Figure 8A*) and PEL embryos (*Figure 8B*). In male PEL embryos, we discovered the Y-linked rDNA in the bridge at only 50% frequency (*Figure 8B*, *Figure 8D*) compared to 95% frequency of the AATAC satellite probe ($p < 0.0001$). Thus, it is the AATAC satellite DNA or an immediately proximal satellite DNA cluster that is in the telophase bridge in male embryos. In contrast, the X-linked rDNA locus in female PEL embryos occurs in the bridge at 85% frequency (*Figure 7D*, *Figure 8B,D*) compared to 60% for the 359 bp repeat ($p < 0.02$). These data suggest that the DNA present in the telophase bridge is more likely to be proximal to the X-rDNA cluster (although not the rDNA itself) than the 359 bp satellite repeats. It is currently unclear whether any of our targeted sequences are responsible for PEL. Nevertheless, our discovery of different frequencies for different probes on the X- and Y-chromosomes implicates discrete loci rather than entire chromosomes underlying mitotic failure, as was discovered for *D. melanogaster-Drosophila simulans* hybrid embryos at nuclear cycles 10–13 (*Ferree and Barbash, 2009*). However, unlike the hybrid case where the 359 bp probe signal appears stretched across the chromatin bridge at these later stage embryos, we observed mostly condensed foci in the bridge of the very first mitosis.

Based on these findings, we conclude that HP1E action is required during sperm development to prime the paternal genome for embryonic chromosome segregation. This priming function is especially critical for faithful segregation of paternal sex chromosomes, which appear to be most vulnerable to HP1E depletion. Even though only a fraction of the paternal genome suffers these

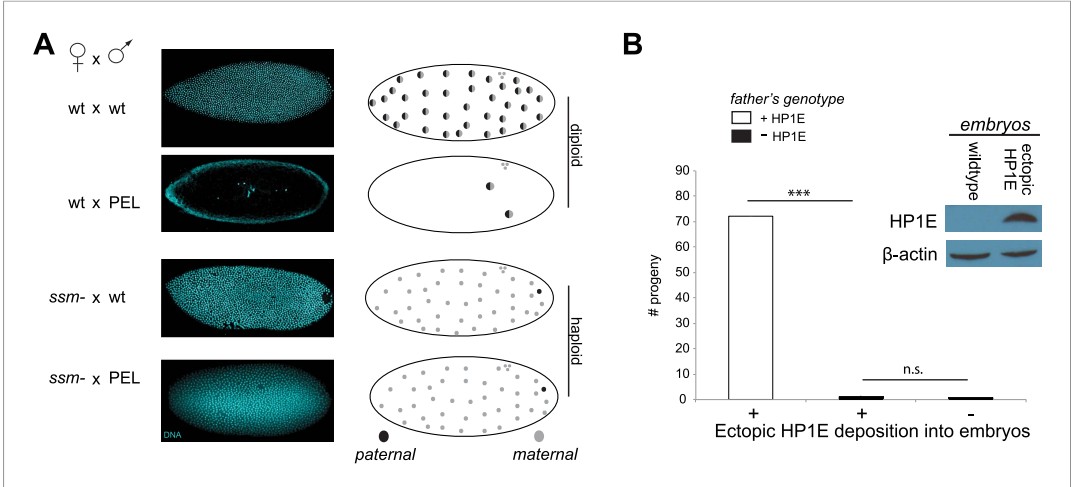

**Figure 6**. Embryonic mitosis can be rescued by excluding paternal chromatin but not by ectopic embryonic deposition of the HP1E protein itself. (**A**) HP1E-knockdown males crossed to wild-type mothers father embryos that undergo early arrest (also see *Figure 3C*). However, both HP1E-depleted and wild-type males crossed to *sesame (ssm)* mothers father maternal haploid embryos that surpass mitotic cycle 12. Black and gray circles refer to the paternal and maternal DNA contributions, respectively, to the zygotic nuclei. Embryos were imaged after fixation and DAPI staining. Please refer to *Figure 6—source data 1*. (**B**) We observe no evidence of rescue when HP1E is deposited ectopically into the egg ('+') prior to fertilization ('***' refers to a p-value < 0.0001 in a Mann–Whitney U test, 'n.s.' = not signficant). Western blot probed with the HP1E antibody shows an absence of native HP1E in embryos of wild-type mothers and HP1E deposition into early embryos of the experimental females. Like wild-type females ('−' on the x-axis), these experimental females ('+' on the x-axis) fail to mother viable progeny when crossed to HP1E-depleted (black bar) males. Please refer to *Figure 6—source data 2*.

The following source data are available for figure 6:

**Source data 1**. Number of embryos generated by *ssm*- females that arrested earlier than cycle 3 ('ARREST') or after cycle 7 ('NOarrest') fathered by wild-type males (24196/TM6) or *HP1E*-depleted males (24196/A5C).

**Source data 2**. Number of progeny generated from crosses between mothers encoding a Gal4 driver alone (*MTD/CyO*) or Gal4 driver plus *UAS-HP1E* construct and males heterozygous ('*HP1E/TM6*') or homozygous (*HP1E-*) for the HP1E mutant chromosome.

consequences, the resulting embryonic mitosis is catastrophic and results in highly penetrant developmental arrest. Thus, paternal contributions laid down during the spermiogenesis program play an essential role in ensuring synchrony of paternal and maternal genomes for the first embryonic mitosis.

## Discussion

Properly coordinated chromosome segregation during virtually all mitotic divisions relies on the function of multiple cell cycle checkpoint proteins (*Lara-Gonzalez et al., 2012*; *Iyer and Rhind, 2013*; *Yasutis and Kozminski, 2013*). No such cell cycle checkpoint proteins have been identified to act in the very first embryonic mitotic cycle (*O'Farrell et al., 2004*), which must nevertheless accomplish the difficult task of synchronizing maternal and paternal chromosomes that were inherited in very different chromatin states. To investigate the paternal contributions that ensure timely participation of the paternal genome in early embryogenesis, we carried out a detailed functional analysis of the testis-restricted *HP1E* gene in *D. melanogaster*. We found that *HP1E* encodes a novel function that ensures paternal genome stability in the embryo. Our cytological and transcriptome analysis revealed that *HP1E* is developmentally restricted within the male germline, where it contributes to heterochromatin integrity. HP1E depletion during sperm development results in a highly penetrant PEL phenotype in which paternal chromosomes, especially the paternal sex chromosomes, fail to condense in synchrony with the maternal chromosomes and ultimately cause mitotic catastrophe. We further showed that the PEL embryonic phenotype could not be rescued by egg-supplied HP1E but could be rescued if the

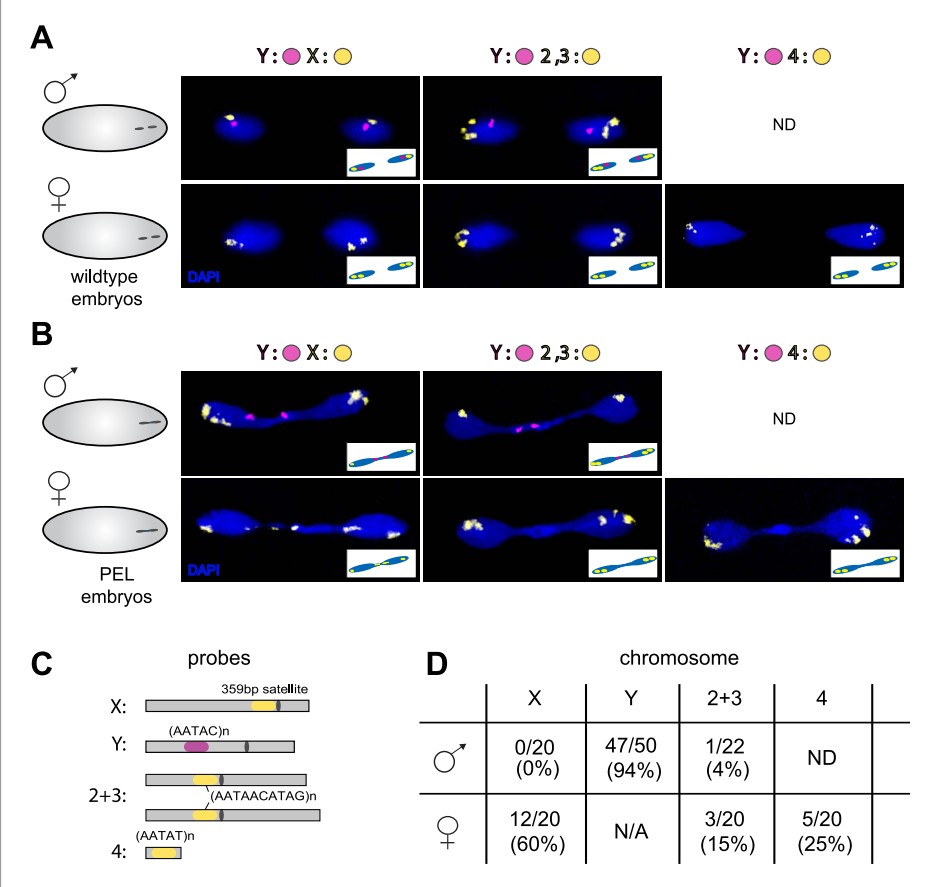

**Figure 7**. Sex chromosomes are especially vulnerable to HP1E depletion in *D. melanogaster*. Representative images of fluorescent in situ hybridization (FISH) analyses of first zygotic telophase in (**A**) wild-type and (**B**) PEL embryos using chromosome-specific satellite probes (**C**), which recognize chromosome-specific repetitive elements (***Dernburg, 2011***). FISH probes against the Y chromosome were tested together with probes against either the chr. X probe (left) or chr. 2 + 3 probe (middle) or chr. 4 probe (right). (**D**) Using at least 20 images per probe pair per sex, we find that sex chromosomes are statistically enriched in the telophase bridge of PEL embryos. Quantification of chromosomal element appearance in the first telophase bridge in male and female embryos (PEL embryos) fathered by HP1E-depleted males. Hybridization of the fourth chromosome probe to the Y chromosome precluded data collection for this probe in male embryos. Data are reported as 'obs/total/(%)' where 'obs' = number of embryos observed with the probe appearing in the telophase bridge, 'total' = total number of embryos sampled per probe, and '%' = obs × 100/total.

paternal DNA was excluded from participating in embryonic mitosis. These observations support a model (*Figure 9A*) under which HP1E acts pre-fertilization to ensure proper chromosome condensation and segregation of paternal chromosomes post-fertilization.

The 'hit and run' priming function clearly distinguishes *HP1E* from all other previously characterized paternal effect lethal genes, which encode proteins that are transmitted to the embryo via sperm (***Browning and Strome, 1996***; ***Fitch and Wakimoto, 1998***; ***Fitch et al., 1998***; ***Loppin et al., 2005b***; ***Smith and Wakimoto, 2007***; ***Gao et al., 2011***; ***Seidel et al., 2011***). These include the *D. melanogaster* paternal chromatin-associated PEL, *k81*, which encodes a protein that persists on paternal telomeres from late spermatogenesis to the first embryonic mitosis (***Dubruille et al., 2010***; ***Gao et al., 2011***). The HP1E-depletion phenotype is instead reminiscent of *Drosophila* fathers infected with *Wolbachia* bacteria crossed to uninfected females (***Serbus et al., 2008***). Embryonic lethality induced by *Wolbachia* testis infection is also caused by a pre-fertilization modification to the paternal genome that results in paternal-maternal chromatin asynchrony and mis-segregation at the very first zygotic mitosis. However, *Wolbachia*-associated PEL results in mis-segregation of the entire paternal

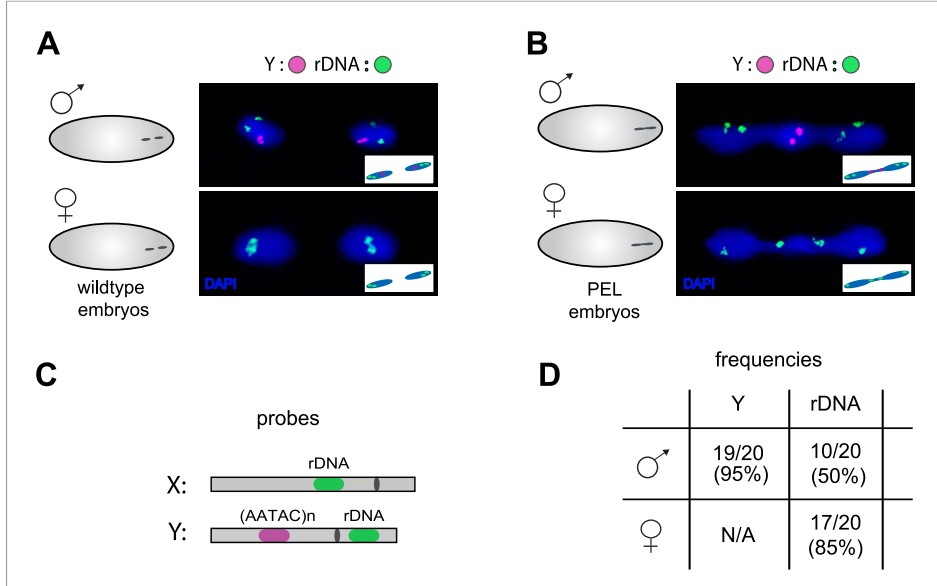

**Figure 8**. Localization of the paternal X- and Y-linked rDNA locus to the telophase bridge in female and male embryos fathered by the HP1E mutant (PEL embryos). Representative images of FISH analyses of first zygotic telophase in (**A**) wild-type and (**B**) PEL embryos using (**C**) probes that recognize the Y-specific satellite AATAC (to determine sex of embryos) and rDNA (the intergenic spacer 'IGS' sequence). (**D**) Quantification of FISH signal in the first telophase bridge in male and female PEL embryos. Data are reported as 'obs/total/(%)' where 'obs' = number of embryos observed with the probe appearing in the telophase bridge, 'total' = total number of embryos sampled per probe, and '(%)' = obs × 100/total.

genome (*Landmann et al., 2009*) rather than just the heterochromatin-rich chromosomes observed in HP1E—PEL (*Figure 5A*, *Figure 7B*). Moreover, the *HP1E* PEL defect is completely independent of *Wolbachia* (we find that PEL phenotype persists for Wolbachia-free males and females). We therefore conclude that *HP1E* supports a novel chromatin requirement to prime paternally inherited genomes for synchronous and successful embryonic mitosis.

How does HP1E ensure timely mitotic entry? It is formally possible that the PEL phenotype is the consequence of a dysregulated spermatid transcriptome that is, up- or down-regulation of a downstream gene. However, our finding that HP1E depletion results in the global up-regulation of heterochromatin-embedded genes, together with our observation that the heterochromatin-rich paternal sex chromosomes are most vulnerable to HP1E depletion, lead us to favor the alternate model that HP1E functions as a canonical HP1 protein during spermiogenesis. Based on antibody localization (*Figure 5—figure supplements 3, 4*) and chromatin bridge morphology (*Figure 5A*), we found no evidence for defects in kinetochore assembly or replication machinery engagement in PEL embryos. Instead, our observation that the lethality phenotype first manifests as decondensed paternal chromosomes relative to maternal chromosomes implicates condensation delay of the heterochromatin-rich sex chromosomes. This delay could be the consequence of incomplete replication (*Landmann et al., 2009*). Indeed, large stretches of uninterrupted heterochromatic DNA, as found on the *Drosophila* sex chromosomes, pose a unique challenge to replication (*Leach et al., 2000*) (*Pryor et al., 1980*; *Collins et al., 2002*). Alternatively, the mitotic delay may be the result of inadequate condensin protein recruitment, which is required for timely resolution of sister chromatids post-replication (*Steffensen et al., 2001*; *Dej et al., 2004*; *Savvidou et al., 2005*; *Cobbe et al., 2006*; *Hirano, 2012*). Previous studies have shown that heterochromatin can also impair chromosome condensation (*Peng and Karpen, 2007*). Timely completion of replication and condensation requires the action of HP1E's closest relative, HP1A, in somatic cells (*Kellum et al., 1995*; *Schwaiger et al., 2010*; *Li et al., 2011*). However, in developing spermatids, HP1A localizes to telomeres (*Dubruille et al., 2010*) rather than broadly to heterochromatin as observed in virtually all other cell types. We posit that HP1E adopts a global, HP1A-like chromatin function during this

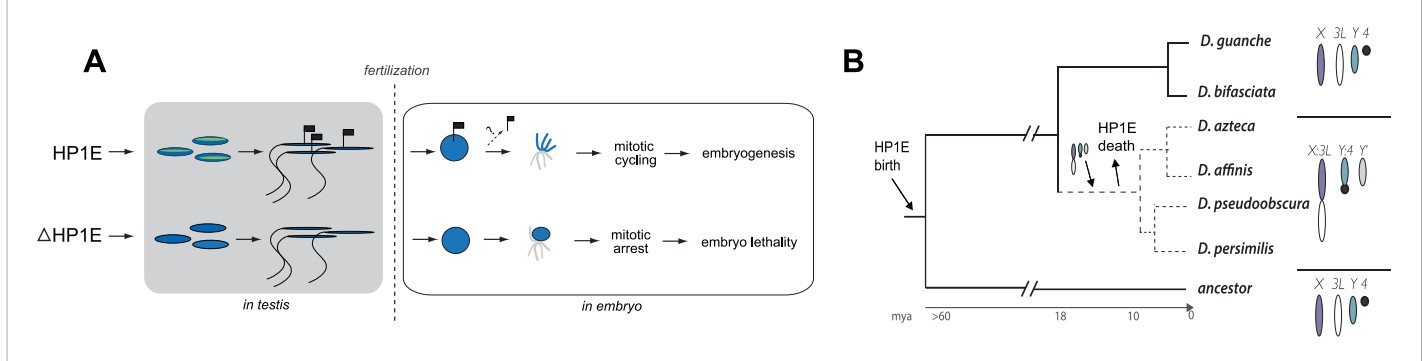

**Figure 9**. Proposed model for HP1E 'hit and run' priming of the paternal genome for timely entry into embryonic mitosis. (**A**) HP1E localization to post-meiotic paternal chromatin directly or indirectly results in an epigenetic mark transferred to the embryo on sperm chromatin. This mark ensures synchronous paternal and maternal entry into the first embryonic mitosis. The absence of HP1E during postmeiotic sperm maturation leads directly or indirectly to the loss of an epigenetic mark (designated by the absence of the flag). Paternal chromatids fail to resolve and mitotic catastrophe ensues. (**B**) The loss of *HP1E* in the obscura group of Drosophila dates to the same 7 million-year long branch as a major karyotype innovation involving the sex chromosomes, including the birth of a neo-Y chromosome (*Carvalho and Clark, 2005*; *Larracuente et al., 2010*; *Levine et al., 2012*). For clarity, only the dynamic subset of the chromosomal elements is presented.

The following figure supplement is available for figure 9:

**Figure supplement 1**. *HP1E* is present for more than 60 million years of Drosophila evolution but was lost at least three times over the Drosophila phylogeny.

highly specialized developmental stage and ensures the recruitment or retention of either replication or condensin proteins that are required post-fertilization.

Previous studies have shown that HP1A is essential for embryo viability (*Eissenberg et al., 1992*). We show here that paternally-acting HP1E is also essential for embryogenesis. Both *HP1A* and *HP1E* evolve under purifying selection (*Levine et al., 2012*). However, unlike HP1A (encoded by *Su(var)205*), *HP1E* has an unusually dynamic evolutionary history. Despite ancient origins, *HP1E* has been recurrently lost over evolutionary time. *HP1E* has been apparently replaced by younger, testis restricted HP1 paralogs on at least two occasions during *Drosophila* evolution (*Levine et al., 2012*) (*Figure 9—figure supplement 1*). Curiously, *Drosophila pseudoobscura* and related species encode neither *HP1E* nor a putative replacement testis-specific HP1 gene. How do we reconcile the paradox of *HP1E* essentiality in *D. melanogaster* with its loss in *D. pseudoobscura*? We previously found that HP1E loss along in *D. pseudoobscura*-related species occurred during the same 7-million evolutionary period as a major sex chromosome rearrangement event (*Levine et al., 2012*), in which the ancestral Y was lost, a neo-Y chromosome was born, and the ancestral X fused to an autosome (*Carvalho and Clark, 2005*; *Larracuente et al., 2010*) (*Figure 8B*). Our finding that the *D. melanogaster* sex chromosomes are especially vulnerable to HP1E depletion, combined with the emergence of novel sex chromosome arrangements along the same narrow branch as *HP1E* pseudogenization (*Figure 9B*), suggests a model under which rearrangements of heterochromatin-rich sex chromosomes in the *obscura* group rendered *HP1E* non-essential. Such karyotypic changes can bring distal heterochromatin into closer proximity to euchromatin and be sufficient to alter heterochromatin packaging (*Spofford, 1976*), replication timing (*Abramov et al., 2005*) or even delete blocks of satellite repeats (*Garagna et al., 1995*). Thus, heterochromatin evolution via chromosomal rearrangements may have obviated maintenance of *HP1E*'s essential heterochromatin function, leading to its degeneration in *D. pseudoobscura*.

Our finding that HP1E is essential in *D. melanogaster* yet lost in the *obscura* group highlights the lineage-restricted essential requirements of chromatin genes. Intriguingly, the only other characterized PEL gene that supports paternal chromatin function in *Drosophila* embryos, *k81*, is similarly lineage-restricted despite being essential for paternal telomere function (*Dubruille et al., 2010*; *Gao et al., 2011*). In contrast, maternally deposited proteins required for paternal chromatin reorganization following fertilization are generally conserved from fly to human (e.g., *Loppin et al.,*

*2005a*; *Konev et al., 2007*; *Delabaere et al., 2014*). This dichotomy is striking. It specifically suggests that even though the essential functions of paternal control of DNA deposition and chromatin remodeling for embryonic mitosis are likely to be conserved in most animals, whereas the identity of those genes is not. PEL chromatin genes like *HP1E* and *k81* thus challenge the dogma that ancient, conserved genes always encode essential conserved functions. Not only can young genes rapidly acquire essential chromatin functions due to dynamic chromatin evolution (*Chen et al., 2010*; *Ross et al., 2013*), but chromatin changes, such as those driven by karyotype evolution, may also drive the extinction of ancient genes encoding once-essential functions (*Drinnenberg et al., 2014*).

# Materials and methods

## Fly transgenics and crossing schemes

To knockdown HP1E expression, we acquired a fly line that encodes a UAS promoter-driven hairpin homologous to the *D. melanogaster HP1E* transcript (line 24196, Vienna Drosophila RNAi Center). We crossed this line to both an *actin5C* Gal4 driver stock (ubiquitous expression, Bloomington #3954) and a *vasa*-driven Gal4 driver (male germline expression, gift of L Jones). Similar data were obtained with both drivers; only *actin5C* driven-RNAi data are presented. We engineered a recoded version of *HP1E* in which all synonymous sites were changed (Genscript Inc.), and PCR-stitched this coding sequence to the *HP1E* UTRs and 1080 bp and 550 bp of 5′ and 3′ flanking noncoding regions, respectively (http://flybase.org). The recoded HP1E transgene was cloned into the pattB vector and engineered into cytolocations 68A4 or 25C6 via injection. The BestGene, Inc. (Chino Hills, CA) carried out this and all other embryo injections using standard procedures. Using genetic crosses, we generated a rescue genotype that encoded one copy of the recoded transgene (cytolocation 68A4) in a background of the Gal4 driver and UAS-driven *HP1E* hairpin. We crossed UASp-driven *HP1E* transgene into the 'MTD' Gal4 driver background (#31777; Bloomington) to overexpress HP1E during oogenesis for deposition into the egg. We generated a 7-base pair lesion in the 5′ region of the HP1E coding sequence using a TAL-effector nuclease (Genetic Services Inc.). We rescued fertility of *HP1E* homozygous mutant fathers by introducing the same transgene inserted at cytolocation 25C6.

## Fertility assays

To assess male fertility, we crossed five 0–5 day old virgin *w1118* females to two 0–5 day old males containing Gal4 driver alone, *HP1E*-hairpin alone, both driver and hairpin, or driver, hairpin and recoded 'rescue'. Parents were discarded after 3 days and progeny counted on day 16. We replicated each cross type four times. To determine if the HP1E-knockdown fathers produced motile sperm, we dissected 10 seminal vesicles in PBS, squashed the tissue between a cover slip and slide, and then examined them under a light microscope. To facilitate sperm imaging, we crossed flies encoding the (*donjuan*) *dj-GFP* construct (*Santel et al., 1997*) into an *HP1E* knockdown background and mounted the male seminal vesicle, the female seminal receptacle to which these males were crossed, and the 5 min-old embryos oviposited by these females. We used a similar scheme to visualize protamine:GFP (*Jayaramaiah Raja and Renkawitz-Pohl, 2005*) in embryos fathered by *HP1E*-knockdown males.

## Assessing embryo arrest, rescue by *sesame*- mothers

To assess the stage of embryonic arrest, we crossed males encoding both the actin5C driver and HP1E hairpin or the driver alone to wild-type *sevelin* females (gift of B Wakimoto). After a 1-hr pre-lay, we collected embryos for 70 min, methanol-fixed each genotype separately (*Rothwell and Sullivan, 2007*), mounted in SlowFade Gold antifade with DAPI (Molecular Probes, Life Technologies Inc., Grand Island, NY), and counted nuclei number/embryo at 20× on a Leica DMI 6000. A Mann–Whitney U test determined significance between the two frequency distributions. To assess embryonic mitotic rescue by *sesame* mothers, we set up the same cross but with virgin *sesame* females (ssm[185b], gift from Kami Ahmad). After a 1-hr pre-lay, *sesame* females oviposited for 45 min followed by 1.5 hr of aging. We then collected, fixed, and counted nuclei as above. For embryos fathered by HP1E knockdown vs wild-type males, we recorded the number of

embryos that underwent early arrest (cycle 3 or earlier) and no arrest (beyond cycle 12). We tested for heterogeneity among the four categories using a Fisher's Exact Test.

## Embryo immunofluorescence

To characterize the progression of paternal chromatin and mitotic machinery leading up to and during the first embryonic mitosis, we conducted immunofluorescence and DAPI staining on wildtype-, *HP1E* knockdown-, or *HP1E/HP1E* fathered embryos that were 0–20 min old. We methanol-fixed embryos (as above) and then rehydrated in PBS plus a drop of PBS + 0.1% Triton. Next, we permeablized in PBS + 1% Triton for 30 min at room temperature. Embryos were blocked in the StartBlock reagent (Thermo Scientific, Waltham, MA) for 90 min at 4°C. We replaced the block with the primary antibody diluted in StartBlock and incubated overnight at 4°C. We then washed embryos in StartBlock for 1 hr following by a 2-hr room temperature incubation in a secondary antibody diluted in StartBlock. After washing embryos for 1 hr in PBS + 0.1% Triton, we mounted them as described above. Primary antibody dilutions were the following: anti-AcH4 (Millipore, Billerica, MA; 1:1000), anti-alpha tubulin (Serotec, Kidlington, UK; 1:250), anti-gamma-tubulin (Sigma–Aldrich, St. Louis, MO), clone GTU-88, 1:1000, anti-Cenp-C (1:5000, gift of C Lehner), anti-PCNA (1:300, gift of P Fisher), and anti-HP1E (1:1000). Alexa-Fluor goat secondary antibodies (Life Technologies) were diluted at 1:1000. We acquired images from the Leica TCS SP5 II confocal microscope with LASAF software and present maximally projected .tif files. Finally, using ImageJ we quantified paternal—maternal metaphase asymmetry by tracing max projected AcH4-staining wild-type and PEL embryos. We measured 'circularity', which is $4\pi \times [Area]/[Perimeter]^2$ and calculated the paternal:maternal ratio.

## HP1E antibody production and western blots

We raised an antibody against HP1E residues CKSLKRGQELNNQYETKAKRLKI and CRILDR-RHYMGQLQYLVKWLDY. Covance Inc. (Princeton, NJ) immunized a single rabbit by injecting it with both peptides over two months. We confirmed HP1E antibody specificity by probing a western blot of nuclear extracts from *Drosophila* S2 cells transfected with a heat-shock inducible, N-terminally Flag-tagged and YFP-tagged HP1E fusion protein plasmids. We designed these constructs using Gateway technology (emb.carnegiescience.edu/labs/murphy/Gateway%20vectors.html) following standard procedures (destination vectors pHFW and pHVW, respectively). We prepared cell lysates by transfecting (FuGENE, Promega, Madison, WI) S2 cells with 2 μg of plasmid DNA and incubating overnight. We transiently induced expression by heat shock (following *Ross et al., 2013*). We washed the recovered cells in PBS and re-suspended in RIPA buffer, sonicated, pelleted, and re-suspended in SDS loading buffer. We probed the membranes with either anti-HP1E (1:500) or anti-Flag ('M2', Sigma–Aldrich, St. Louis, MO; 1:2000) primary antibodies followed by goat anti-rabbit or goat anti-mouse IgG-HRP (Santa Cruz Biotechnologies Inc., Dallas, TX). Lysates for the western blot confirming deposition of ectopically-expressed HP1E into embryos were prepared from 0–40 min embryos laid by females encoding the MTD driver and either UASp-HP1E transgene (described above) or a balancer chromosome. We flash froze the embryos followed by grinding with a glass pipette in SDS loading buffer and boiling at 95°C for 5 min. We probed the membranes with anti-HP1E and anti-beta actin (Abcam, Cambridge, UK, both at 1:5000).

## Testis immunofluorescence

To assess HP1E localization in testis, we used the HP1E antibody or generated transgenic flies encoding an N-terminal Flag- or YFP-tag fused to HP1E, flanked by the native promoter and 5′ and 3′ regions. To cytologically characterize HP1E localization without antibody staining, we fixed testis from 2–5 day old YFP-HP1E males in 4% paraformaldehyde (PFA) and mounted in SlowFade Gold antifade with DAPI. For immunofluorescence, we fixed Flag-tagged and untagged testis in 4% PFA in periodate-lysine-paraformaldehyde (PLP) for 1 hr followed by 30 min in PBS + 0.3% Triton, 0.3% sodium deoxycholate. After a 10 min wash in PBS + 0.1% Triton, we blocked testis for 30 min in PBS + 0.1% Triton + 3% BSA followed by standard IF procedures using the following dilutions: anti-Flag ('M2', Sigma–Aldrich, St. Louis, MO; 1:2500), anti-HP1E (1:1000), and Alexa Fluor secondaries goat anti-mouse 488 (1:1000) and goat anti-rabbit (1:1000). We

acquired images from the Leica TCS SP5 II confocal microscope with LASAF software and present maximally projected tagged image files (tifs).

## RNA-seq methods

We prepared RNA from testis dissected from three biological replicates per genotype representing three independent crosses of males heterozygous for the UAS promoter-driven hairpin homologous to the *D. melanogaster HP1E* transcript and virgin females homozygous for vasaGAL4 inserted on chromosome II. The FHCRC Shared Resources Genomics Core prepared six libraries using Illumina TruSeq Sample Prep Kit v2. We performed image analysis and base calling with Illumina's RTA v1.13 software and demulitplexed with Illumina's CASAVA v1.8.2. We aligned reads to BDGP5r66 using TopHat v1.4.0 and converted files to sam format using samtools v0.1.18. We used htseq-count v0.5.3 to generate counts/gene and removed genes that had 0 counts across all samples or less than 1 count/million in at least three samples. This culling resulted in 11,051 genes. We identified differentially expressed genes using edgeR v2.6 and tested for significant enrichment of up-regulated heterochromatin-embedded genes using binomial probability. We annotated heterochromatin-embedded genes using the *D. melanogaster* Release 5.

## Fluorescent in situ hybridization

To determine if participation in chromatin bridging was chromosome-specific, we designed Cy3 or Cy5 conjugated probes (IDT) for in situ hybridization following (*Dernburg, 2011*): X (359 bp satellite), Y (AATAC)n, 2L + 3L (AATAACATAG)n, 4 (AATAT)n and IGS (GTATGTGTTCATAT-GATTTTGGCAATTATA, ATATTCCCATATTCTCTAAGTATTATAGAG, designed by P Ferree). We co-hybridized two probes using the following conjugated tags and annealing temperatures: 32°C for Y-Cy5 + 2L/3L-Cy3 and Y + X-Cy3, 23°C for Y + 4-Cy3 and 30°C for Y-Cy5 + IGS-Cy3. For the FISH experiments, we modified the above embryo fixation protocol by replacing methanol:heptane with 3.7% paraformaldehyde:heptane.

## Acknowledgements

The authors thank B Wakimoto, L Jones, K Ahmad, P Fisher, and Y Yamashita for reagents, J Vasquez for microscopy support, and A Larracuente for the IGS probe sequences. We thank C Peichel, P Ferree, K Ahmad, members of the Malik lab and the reviewers for their comments and valuable discussions. This work was supported by an NIH K99/R00 Pathway to Independence Fellowship GM107351 to MTL and grants from the Mathers Foundation and NIH R01 GM74108 to HSM. HSM is an investigator of the Howard Hughes Medical Institute.

## Additional information

### Funding

| Funder | Grant reference | Author |
| --- | --- | --- |
| Howard Hughes Medical Institute (HHMI) | HHMI Investigator | Harmit S Malik |
| National Institute of General Medical Sciences (NIGMS) | K99/R00 GM107351 | Mia T Levine |
| National Institute of General Medical Sciences (NIGMS) | R01 GM74108 | Harmit S Malik |
| G Harold and Leila Y. Mathers Foundation | Project Grant | Harmit S Malik |

The funders had no role in study design, data collection and interpretation, or the decision to submit the work for publication.

## Author contributions
MTL, Conceived and designed the study, performed or participated in all of the experiments, analyzed the data and wrote the paper; HMVW, Performed experiments (fly crosses, embryo collections and FISH), analyzed the data and helped edit the paper; HSM, Conceived and designed the study, helped with analysis of the data, and wrote the paper

## Author ORCIDs
Harmit S Malik, http://orcid.org/0000-0001-6005-0016

# Additional files

## Major datasets
The following datasets were generated:

| Author(s) | Year | Dataset title | Dataset ID and/or URL | Database, license, and accessibility information |
|---|---|---|---|---|
| Levine MT, Wende HV, Malik HS | 2015 | Gene expression change upon knockdown of HP1E in Drosophila melanogaster testis | http://www.ncbi.nlm.nih.gov/sra/?term=SRX1045369 | Publicly available at the NCBI Short Read Archive (Accession no: SRX1045369). |
| Levine MT, Wende HV, Malik HS | 2015 | Gene expression change upon knockdown of HP1E in Drosophila melanogaster testis | http://www.ncbi.nlm.nih.gov/sra/?term=SRX1045369 | Publicly available at the NCBI Short Read Archive (Accession no: SRX1045369). |

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
