## [Decision Letter]

Thank you for sending your work entitled “Mitotic fidelity requires transgenerational action of a testis-restricted HP1” for consideration at *eLife*. Your article has been favorably evaluated by K VijayRaghavan (Senior editor), a Reviewing editor, and three reviewers.

The following individuals responsible for the peer review of your submission have agreed to reveal their identity: Asifa Akhtar (Reviewing editor), William Theurkauf and Daniel Barbash (peer reviewers). A further reviewer remains anonymous.

Summary:

The reviewers have provided their individual assessments and also discussed in detail the findings presented in your manuscript. The reviewers are of the opinion that it is an interesting story with elegant genetics to describe the role of HPIE as a paternal effect mutation required for segregation of paternally-derived chromosomes. However, additional work is required to strengthen your claims so that the manuscript is a strong candidate for publication in *eLife*.

Essential revisions:

Detailed reviewers’ comments are included below, but the consensus among the reviewers was there are at least two major points we would like you to address upon revision.

1) The authors need to strengthen their claim about the “heterochromatin function” of HP1E. RNA seq is interesting but more direct evidence would be needed. Ideally ChIP-seq from this particular stage will be desirable however, if this is not feasible then at least ChIP-qPCR for a selected gene set will be important to provide more direct evidence for HPIE action on heterochromatin.

2) The authors also need a better characterization of the mis-segregation phenotype. As pointed out in one of the reviewer's comments, it will be very helpful if the authors used additional probes such as rDNA and a few euchromatic loci and centromeres and compared their phenotype with other reported work at later stages.

Reviewer #1:

Only a few minor comments.

1) Related to the data that show narrow expression window, yet causing derepression of heterochromatin genes in mutant: What kind of chromatin state do the authors think HP1E is conferring to sperm chromatin that both explains derepression of heterochromatin genes during spermatogenesis AND defective chromatin decondensation and segregation of paternal chromosomes in zygotes?

2) Related to their conclusion that HP1E is depleted from mature sperm chromatin, any discussion on the possibly very low amount of it marking certain genomic location, similar to a finding that histone is not completely removed from mammalian sperm chromatin? (http://www.ncbi.nlm.nih.gov/pubmed/19525931)

Reviewer #2:

This report shows that knock down and chromosomal mutations that disrupt the *Drosophila* testes specific HP1, HP1E, lead to defects in remodeling of the male pronucleus and segregation of the sex chromosomes during the early cleavage stage embryonic divisions. Western blotting and IF data indicate that HP1E protein localizes to nuclei during spermatogenesis, but is not incorporated into mature sperm or transferred to the embryo. The authors also show that maternal expression of HP1E does not rescue the PEL phenotype, and use a very elegant genetic experiment to show that the embryonic mitotic defects are specifically due to failure to segregate the paternal chromosomes. Based on these observations, the authors propose that HP1E establishes an epigenetic mark that is essential to reorganization of the male pronucleus. The data are consistent with this model, but HP1E mutations alter expression of over 700 genes, and it is also possible that one or more of these genes establish an inherited modification or encode a chromatin protein that is required for reorganization and segregation of the male pronucleus. While directly testing this hypothesis is difficult, this possibility needs to be considered /discussed. Nonetheless, the experiments are well executed, clearly presented, and genetically define a novel paternal function in pronuclear reorganization. The additional experiments suggested below are not essential, but would strengthen the story and could provide mechanistic insights:

1) HP1E depletion leads to mis-regulation of heterochromatin genes and segregation defects predominantly for sex chromosomes during embryonic mitosis in next generation. Does HP1E preferentially associate with the sex chromosomes during spermatogenesis? ChIP sequencing would be the best way to test this. Is this technically challenging in fly testes? If not, the authors appear to have the reagents needed to perform the experiment, and it could provide important insight into HP1E function.

2) HP1E is likely to bind H3K9-me3, and co-localization of HP1E and K9-me3 would indirectly test this. HP1A recruits the methyl transferase that generates this binding site, leading to heterochromatin spreading. HP1E may have a similar function. It would be interesting to determine if H3K9me3 localization is altered in HP1E mutants. This could be done by IF, but ChIPseq would be better.

3) As noted above, the RNAseq data show that many genes change their expression in HP1E mutants. Do any these genes encode chromatin modifying proteins, or proteins that directly associate with chromatin?

Reviewer #3:

Levine et al. have characterized the gene HP1E, discovering that it is a paternal effect mutation required for segregation of paternally-derived chromosomes. The combination of the FISH, anti-AcH4, and sesame mutant experiments are particularly strong with regards to HP1E's requirement for paternal chromosome segregation in the first embryonic mitosis. The authors further establish that HP1E is present and thus likely active only in developing sperm, with a mechanistic understanding of its role in chromosome segregation awaiting further study. The authors suggest that HP1E is a heterochromatin factor but this is based on mostly indirect evidence.

Major points:

1) I'm not very familiar with the spermatid stage being studied but take the authors' point from cited refs that heterochromatin cannot be identified with HP1A, H3MeK9, etc. at that stage. So the inability to directly examine HP1E chromatin localization is an acceptable limitation of the system. Localization in other tissue types upon ectopic expression obviously has caveats but would be useful to show if available.

The RNA-seq analysis is presented as an indirect alternative to determine whether HP1E is a heterochromatin protein, but more could potentially be gained from this, and the analyses clarified. How are heterochromatin genes defined in this study? What proportion of them are misregulated—all misregulated may be upregulated, but out of what total? Are they equally affected on all chromosome arms—this might give some insight into the apparent specificity of the segregation defect? How do such results compare to euchromatic genes? Chromosome 4 would be of particular interest due to its unusual heterochromatic state.

Analysis of repetitive sequences could also be informative for addressing whether HP1E is a general or specific chromatin regulator. There's little doubt that satellite sequences are not proportionally represented in Illumina data, especially RNA-Seq, but we have had some success with Hmr and Lhr in mapping reads to satellites, with the results partially correlating with cytological data.

2) Relating to the issue of whether HP1E is a heterochromatin protein, and its degree of specificity, the mis-segregation phenotypes in Figure 7 look very different from what we've seen in Ferree's work at later embryonic stages, where mis-segregating sequences like 359 go all the way across the anaphase bridges. Here with HP1E mutant the probes are forming clusters that don't look any more stretched out compared to normal segregation, e.g., when comparing the presumed maternal and paternal X's in the female embryo. So most of the stretched material in the bridge is unidentified. It would be informative to look at additional probes such as rDNA and a few euchromatic loci. Were centromeres looked at (may have missed)? Figure 7 (and others) are also lacking wild type controls. Here they are really essential in order to interpret the mutant phenotypes.

3) HP1E is firmly established here as a paternal effect mutant. “Transgenerational” is widely understood to refer to mutant phenotypes that are transmitted across generations, and does not apply here.

---

## [Author Response]

*1) The authors need to strengthen their claim about the “heterochromatin function” of HP1E. RNA seq is interesting but more direct evidence would be needed. Ideally ChIP-seq from this particular stage will be desirable however, if this is not feasible then at least ChIP-qPCR for a selected gene set will be important to provide more direct evidence for HPIE action on heterochromatin*.

We absolutely agree with the reviewers that our claim would be substantially bolstered by a ChIP-seq analysis to support our inference that HP1E is a heterochromatin protein. We made a number of unsuccessful attempts prior to our original submission to obtain reliable ChIP-seq data on HP1E in *Drosophila* testes. Unfortunately, ChIP on *Drosophila* testis is notoriously challenging and only a few groups have reported a successful chIP-seq experiment. In those few cases, research groups have either targeted a highly abundant protein or histone mark or have taken advantage of a *Drosophila* mutant, *bam*, which enriches the testis for the undifferentiated cell type of interest. Unfortunately, neither of these is applicable to the study of HP1E, which is expressed during spermiogenesis only and is not abundant.

After receiving the reviewer request, we partnered with the Henikoff lab here at the Fred Hutch to attempt chIP using a vetted anti-Flag reagent on *D. melanogaster* testis encoding a Flag:HP1E transgene. Unfortunately, we were unsuccessful despite multiple attempts that took advantage of increasing amounts of starting material. Without reliable recovery of DNA, even chIP-qPCR is hard to validate.

As an alternate strategy, we also considered conducting ChIP-seq on HP1E transfected into S2 or Kc cells. However, we have found that (unlike HP1A, HP1B, HP1C, and HP1D which we have published on previously) HP1E fails to localize to chromatin in cell culture. This suggests the possibility of a testis-restricted cofactor required for HP1E’s chromatin localization. In any case, we have concluded that we cannot use cell lines for this experiment.

We have now turned to a newly developed ‘CHEC-seq’ strategy developed by the Henikoff lab that requires construction of transgenic flies encoding an MNase fusion to HP1E (works similar to the Dam-ID strategy developed by Bas van Steensel). Unfortunately, we are only at the beginning of this experiment, which will require at least another year of development and validation.

Given these technical limitations, we have responded to this concern by moderating the language in the first section, stating now that “…in the absence of direct evidence of heterochromatin localization via cytology or ChIP-seq, we can only tentatively conclude that HP1E acts directly on this genome compartment” (subsection headed “HP1E encodes a spermiogenesis-restricted chromatin protein”). We note that none of our main conclusions depend on HP1E’s direct action at heterochromatin. However, our finding that HP1E loss leads to chromatin bridges, with at least heterochromatic satellites on Y localizing to this bridge, is highly consistent with HP1E action on heterochromatin.

*2) The authors also need a better characterization of the mis-segregation phenotype. As pointed out in one of the reviewer's comments, it will be very helpful if the authors used additional probes such as rDNA and a few euchromatic loci and centromeres and compared their phenotype with other reported work at later stages*.

While our primary goal for the FISH experiments was to assess the possibility of chromosome specific enrichment in a chromatin bridge, we agree that identifying the offending DNA segment would be extremely informative. We especially agree that testing the *rDNA* locus is important given that it is a rare repetitive locus uniquely shared by the *D. melanogaster* X and Y chromosomes. This locus is also technically feasible to investigate in the very early embryo, which is notoriously challenging developmental stage in which to image weak signal (as is typical from non-repetitive targets). In new experiments since our original submission, we have now designed two probes that target the IGS (repetitive) region of the *rDNA* locus. We hybridized these probes in combination to 20 male and 20 female *HP1E* mutant-fathered early embryos. The findings from this new experiment are now reported in the new Figure 8 and in the Results section (subsection headed “Heterochromatin-rich paternal sex chromosomes are especially vulnerable to HP1E depletion”). These new data offer an additional contribution to the original FISH table by providing information about the relative likelihood of rDNA versus Y- and X-chromosomal satellites to appear in the telophase bridge. For instance, we found that the Y-linked AATAC repeat is present at the bridge nearly 95% of the time whereas the Y-linked rDNA is present only 50% of the time. In contrast, in female embryos, the rDNA is present in the bridge more often than the X-linked 359 bp satellite (85% versus 60% respectively). We have also now contextualized these results with respect to the similar discovery of Ferree and Barbash (PLOS Biology 2009).

In contrast to rDNA, non-repetitive euchromatic loci are also not reliably probed via FISH in the first anaphase/telophase. Moreover, our discovery that euchromatin-encoding chromosomes are very rarely found in the bridge (6%) and of the dominant presence of the entirely heterochromatic Y chromosome in the bridge, leads to an a priori reason to hypothesize that euchromatic loci are the source of vulnerability.

Reviewer #1:

*Only a few minor comments*.

1) Related to the data that show narrow expression window, yet causing derepression of heterochromatin genes in mutant: What kind of chromatin state do the authors think HP1E is conferring to sperm chromatin that both explains derepression of heterochromatin genes during spermatogenesis AND defective chromatin decondensation and segregation of paternal chromosomes in zygotes?

Chromatin packaging at this late stage of spermatogenesis and post-fertilization/ pre-embryonic mitosis is poorly characterized, particularly in *Drosophila*. This dearth of information disables us from proposing a more specific model than the one presented in the paper. We do note that heterochromatin packaging by HP1 proteins in mitosis is required for both transcriptional silencing and high-fidelity DNA repair in heterochromatin (work from Gary Karpen’s lab that we cite).

*2) Related to their conclusion that HP1E is depleted from mature sperm chromatin, any discussion on the possibly very low amount of it marking certain genomic location, similar to a finding that histone is not completely removed from mammalian sperm chromatin? (**http://www.ncbi.nlm.nih.gov/pubmed/19525931**)*

We agree with the reviewer that small amounts of HP1E could be present in either tissue type but would not be detectable by the assays we have performed. We tried to address this possibility by attempting to rescue embryo viability by ectopic HP1E deposition into the egg via overexpression during oogenesis. However, this ectopic expression would not be able to overcome any defect associated with loss of HP1E retention of particular loci in paternal chromatin. Again, the absence of data on histone retention in *Drosophila* sperm makes speculation beyond our dataset difficult.

Reviewer #2:

*This report shows that knock down and chromosomal mutations that disrupt the* Drosophila *testes specific HP1, HP1E, lead to defects in remodeling of the male pronucleus and segregation of the sex chromosomes during the early cleavage stage embryonic divisions. Western blotting and IF data indicate that HP1E protein localizes to nuclei during spermatogenesis, but is not incorporated into mature sperm or transferred to the embryo. The authors also show that maternal expression of HP1E does not rescue the PEL phenotype, and use a very elegant genetic experiment to show that the embryonic mitotic defects are specifically due to failure to segregate the paternal chromosomes. Based on these observations, the authors propose that HP1E establishes an epigenetic mark that is essential to reorganization of the male pronucleus. The data are consistent with this model, but HP1E mutations alter expression of over 700 genes, and it is also possible that one or more of these genes establish an inherited modification or encode a chromatin protein that is required for reorganization and segregation of the male pronucleus. While directly testing this hypothesis is difficult, this possibility needs to be considered /discussed*.

We agree with the reviewer. We had indeed failed to adequately highlight this potential mechanism of action and so have added a new sentence raising this possibility (in the subsection headed “HP1E encodes a spermiogenesis-restricted chromatin protein”).

*Nonetheless, the experiments are well executed, clearly presented, and genetically define a novel paternal function in pronuclear reorganization. The additional experiments suggested below are not essential, but would strengthen the story and could provide mechanistic insights*:

1) HP1E depletion leads to mis-regulation of heterochromatin genes and segregation defects predominantly for sex chromosomes during embryonic mitosis in next generation. Does HP1E preferentially associate with the sex chromosomes during spermatogenesis?

We wondered the same thing and conducted immuno-FISH on testis to test this hypothesis but did not see co-localization. Instead, we found that HP1E mark on the developing spermatids is quite diffuse whereas the FISH signals are not.

*ChIP sequencing would be the best way to test this. Is this technically challenging in fly testes? If not, the authors appear to have the reagents needed to perform the experiment, and it could provide important insight into HP1E function*.

As we indicated earlier in our response, we have found chIP-seq on HP1E in testes to be extremely challenging (we developed the HP1E-FLAG fusion reagent in part to facilitate these analyses). We have devoted the bulk of our time since receiving the reviews, and even time prior to initial submission, to get HP1E chIP-seq to work. However, our DNA yield is too poor to be reliable even for chIP-PCR analyses. Despite hundreds of testis pairs and use of an anti-Flag antibody used with recurrent success in the Henikoff lab, we were unable to recover detectable amounts of DNA from the IP. We speculate that our failure to successfully perform chIP against HP1E:Flag is related to the small number of cells in testis within which HP1E localizes.

*2) HP1E is likely to bind H3K9-me3, and co-localization of HP1E and K9-me3 would indirectly test this. HP1A recruits the methyl transferase that generates this binding site, leading to heterochromatin spreading. HP1E may have a similar function. It would be interesting to determine if H3K9me3 localization is altered in HP1E mutants. This could be done by IF, but ChIPseq would be better*.

We agree that a perturbation of H3K9me3 in HP1E mutant testis would be extremely interesting; however, chromatin organization in late spermatogenesis in *Drosophila* is poorly characterized. We would first need to rigorously describe the distribution of the H3K9me3 mark in wildtype testes and then compare it to the HP1E-knockout case. Moreover, since only a small subset of cells in the testes express HP1E, there is a high likelihood that any effect of HP1E depletion on H3K9me3 would be masked by heterogeneous testis cell types.

3) As noted above, the RNAseq data show that many genes change their expression in HP1E mutants. Do any these genes encode chromatin modifying proteins, or proteins that directly associate with chromatin?

This question was one of the first we investigated after analyzing the RNA-seq data. We did not highlight this because very few genes emerged that are known to encode chromatin modifying proteins or chromosome-bound proteins. The few that came out of our go analysis were all uncharacterized “CGs”. We now explicitly refer to this result (please see the subsection “HP1E encodes a spermiogenesis-restricted chromatin protein”).

Reviewer #3:

*Major points*:

*1) I'm not very familiar with the spermatid stage being studied but take the authors' point from cited refs that heterochromatin cannot be identified with HP1A, H3MeK9, etc. at that stage. So the inability to directly examine HP1E chromatin localization is an acceptable limitation of the system. Localization in other tissue types upon ectopic expression obviously has caveats but would be useful to show if available*.

We agree that cytological data establishing HP1E heterochromatin localization even using ectopic overexpression would be ideal. However, we previously discovered HP1E fails to localize to chromatin when overexpressed in tissue culture cells (unlike the female germline HP1D, HP1D/Rhino). The striking difference between somatic and germline localization complicates interpretation of HP1E ectopic overexpression experiments.

The RNA-seq analysis is presented as an indirect alternative to determine whether HP1E is a heterochromatin protein, but more could potentially be gained from this, and the analyses clarified. How are heterochromatin genes defined in this study?

Heterochromatin genes are defined by their chromosomal locations according to the *Drosophila melanogaster* Release 5 cytolocations. We now explicitly refer to the annotation of these heterochromatin-embedded genes (in the subsection headed “RNA-seq Methods”). Also in response to Reviewer 2’s request, we now give more details about the genes that are overexpressed (unfortunately most of them are uncharacterized).

*What proportion of them are misregulated—all misregulated may be upregulated*, *but out of what total?*

1/3 of all heterochromatin-embedded genes are significantly misregulated at a p-value of 0.05 and 1/4 at 0.02. Possibly even more compelling was our finding that over 90% of all heterochromatin-embedded genes are upregulated independent of significance level ([Supplementary-material SD1-data]).

*Are they equally affected on all chromosome arms—this might give some insight into the apparent specificity of the segregation defect? How do such results compare to euchromatic genes? Chromosome 4 would be of particular interest due to its unusual heterochromatic state*.

This is a good idea. Unfortunately, we did not detect any significant chromosome effects, including for 4^th^ chromosomes genes, but our power is severely limited by the low numbers of heterochromatin-embedded genes annotated per chromosome.

*Analysis of repetitive sequences could also be informative for addressing whether HP1E is a general or specific chromatin regulator. There's little doubt that satellite sequences are not proportionally represented in Illumina data, especially RNA-Seq, but we have had some success with Hmr and Lhr in mapping reads to satellites, with the results partially correlating with cytological data*.

We did map our reads to repetitive elements (using the repeat masker file, which also contains satellite repeats) but did not find any compelling patterns of up- (or down-) regulation. Only a single element (GTWIN_LTR) exhibited a log -fold change > 2 (2.1), and it was down rather than up in the mutant testis.

*2) Relating to the issue of whether HP1E is a heterochromatin protein, and its degree of specificity, the mis-segregation phenotypes in Figyre 7 look very different from what we've seen in Ferree's work at later embryonic stages, where mis-segregating sequences like 359 go all the way across the anaphase bridges. Here with HP1E mutant the probes are forming clusters that don't look any more stretched out compared to normal segregation, e.g., when comparing the presumed maternal and paternal X's in the female embryo. So most of the stretched material in the bridge is unidentified. It would be informative to look at additional probes such as rDNA and a few euchromatic loci. Were centromeres looked at (may have missed)?*
Figure 7
*(and others) are also lacking wild type controls. Here they are really essential in order to interpret the mutant phenotypes*.

We agree with the reviewer’s comment that that the X chromosome bridge phenotype is quite distinct from the Ferree and Barbash publication. We now address this difference in the Results section (“Heterochromatin-rich paternal sex chromosomes are especially vulnerable to HP1E depletion”). We have also now conducted a new FISH experiment directed at the rDNA (see above and Results section). We also did investigate the centromeres (via CenpC staining); this data appears in Figure 5—figure supplement 4. We have now added the wildtype controls for FISH data in a new Figure 7 panel A and for the rDNA staining in Figure 8 panel A.

*3) HP1E is firmly established here as a paternal effect mutant. “Transgenerational” is widely understood to refer to mutant phenotypes that are transmitted across generations, and does not apply here*.

We think the term transgenerational action accurately refers to the fact that the phenotype of HP1E loss is transmitted to the next generation even though the protein is not obviously inherited. However, we are open to changing the terminology if there was consensus about this term.